

# TECHNICAL NOTE: Coupling infrared gas analysis and cavity ring down spectroscopy for autonomous, high temporal resolution measurements of DIC and δ13C-DIC

Mitchell Call[1,3], Kai G. Schulz[2], Matheus C. Carvalho[2], Isaac R. Santos[3] and Damien T. Maher[1,3]

[1]School of Environment, Science and Engineering, Southern Cross University, Lismore, NSW, 2480, Australia
[2]Centre for Coastal Biogeochemistry, School of Environment, Science and Engineering, Southern Cross University, Lismore, NSW, 2480, Australia
[3]National Marine Science Centre, School of Environment, Science and Engineering, Southern Cross University, Coffs Harbour, NSW, 2450, Australia

*Correspondence to:* Mitchell Call (m.call.10@student.scu.edu.au)

**Abstract.** A new approach to autonomously determine concentrations of dissolved inorganic carbon (DIC) and its carbon stable isotope ratio (δ13C-DIC) at high temporal resolution is presented. The simple method requires no customised design. Instead it uses two commercially available instruments currently used in aquatic carbon research. An inorganic carbon analyser utilising non-dispersive infrared detection (NDIR) is coupled to a Cavity Ring-down Spectrometer (CRDS) to determine DIC and δ13C-DIC based on the liberated $CO_2$ from acidified aliquots of water. Using a small sample volume of 2 ml, the precision and accuracy of the new method was comparable to standard isotope ratio mass spectrometry (IRMS) methods. The system achieved a sampling resolution of 16 mins, with a DIC precision of ± 1.5 to 2 µmol kg⁻¹ and δ13C-DIC precision of ± 0.14 ‰ for concentrations spanning 1000 to 3600 µmol kg⁻¹. Accuracy of 0.1 ± 0.06 ‰ based on DIC concentrations ranging from 2000 µmol kg⁻¹ to 2230 µmol kg⁻¹ was achieved during a laboratory-based algal bloom experiment. The high precision data that can be autonomously obtained by the system should enable complex carbonate system questions to be explored in aquatic sciences using high temporal resolution observations.

**Keywords.** Dissolved inorganic carbon (DIC), stable isotopes, carbonate chemistry, laser spectroscopy, keeling plot.

## 1 Introduction

Dissolved inorganic carbon (DIC) is an important component of the evolving global carbon cycle, with ~ 26 % of yearly anthropogenic carbon dioxide ($CO_2$) emissions stored as DIC in the global ocean (Le Quéré et al., 2015). This influx of carbon to the oceans has resulted in increased field-monitoring (Sabine et al., 2010), laboratory-based experiments of how changing seawater carbonate chemistry effects biological process (Gattuso and Hansson, 2011), as well as the development of new measurement technologies (Byrne, 2014;Martz et al., 2015). Currently, the spatial and temporal coverage of paired DIC and carbon stable isotope ratio (δ13C-DIC) measurements is poor (Becker et al., 2016). However, greater interest is being placed on the coupled high-resolution measurement of DIC and δ13C-DIC as it can provide insights into the processes controlling DIC concentrations, helping elucidate flows of carbon within and between reservoirs (Bass et al., 2014b).





Whilst a variety of methods to autonomously measure DIC concentrations have been developed (Bandstra et al., 2006;Fassbender et al., 2015;Huang et al., 2015;Liu et al., 2013) the conventional method for determining $\delta^{13}$C-DIC requires discrete samples to be collected and stored prior to acidification and analysis in a laboratory by isotope ratio mass spectrometry (IRMS). IRMS analysis offers high precision, however, the collection, handling and preservation of discrete samples may introduce sampling artefacts and reduce accuracy (Li and Liu, 2011;Taipale and Sonninen, 2009). Furthermore, the laborious process limits sampling frequency, resulting in low temporal and spatial coverage of coupled DIC and $\delta^{13}$C-DIC measurements.

High resolution, field-based measurement of $CO_2$ and its carbon stable isotope value ($\delta^{13}$C-$CO_2$) is now achievable via laser spectroscopy systems such as Cavity Ring-Down Spectrometers (CRDS) (Crosson, 2008) and Off-Axis Integrated Cavity Output Spectroscopy (OA-ICOS) (Baer et al., 2002), with precision and accuracy comparable to laboratory-based IRMS (Vogel et al., 2013;Berryman et al., 2011;Midwood and Millard, 2011). The use of CRDS in the aquatic environment is becoming more prevalent with CRDS successfully coupled to air-water equilibrators for on-site, high-resolution measurement of dissolved $CO_2$ and its isotopic composition (Maher et al., 2013b;Becker et al., 2012). Recently, Bass et al. (2012) coupled a CRDS to a customised acidification interface utilising expanded polytetrafluoroethylene (ePTFE) tubing to measure in-situ concentrations of DIC and its $\delta^{13}$C-DIC. The permeable membrane based equilibration system autonomously measured DIC and $\delta^{13}$C-DIC at 15 minute intervals to a precision of $\pm$ 10 µmol kg$^{-1}$ and $\pm$ 0.2 ‰ respectively, and has been shown to be sufficient for identifying spatial and short-term temporal variability in DIC concentrations in a variety of aquatic systems (Bass et al., 2014a;Bass et al., 2014b;Bass et al., 2013). However, a precision of ~2 µmol kg$^{-1}$ or better is required in order to assess other processes such as long-term anthropogenic-induced changes to oceanic carbon chemistry (Newton et al., 2014) or in laboratory-based experiments when trying to detect small changes in DIC.

This paper presents an alternative approach to autonomously determine concentrations of DIC and $\delta^{13}$C-DIC. The simple method does not require the need to design or replicate a customised system. Instead, it couples two commercially available instruments, an inorganic carbon analyser utilising non-dispersive infrared detection (NDIR) and a CRDS. The system can be automated, is low maintenance, and achieves a sampling resolution of ~ 16 mins. Using only a small sample volume (2 ml), the method achieves high precision and accuracy comparable to traditional IRMS techniques.

## 2 Materials and procedures

### 2.1 Approach

Two commercially available instruments, an Autonomous Infra Red Inorganic Carbon Analyser (AIRICA, Marianda Company, Kiel, Germany), and a CRDS (Picarro G2201-*i*, Picarro Inc., Santa Clara, CA. USA) were coupled to autonomously measure concentrations of DIC and its carbon stable isotope ratio ($\delta^{13}$C-DIC) based on the $CO_2$ extracted from acidified samples (Fig. 1). These instruments offer high precision and accuracy, and are currently used in aquatic carbon research. To test the response of the method, here-in termed AIRICA-CRDS, laboratory-based experiments on precision were conducted as well as a mesocosm experiment simulating an algal bloom in coastal waters.





### 2.2 Instrumentation

The AIRICA system determines DIC to within ± 1.5 to 2 µmol kg$^{-1}$ (0.1 %) based on the NDIR gas analysis method (Goyet and Snover, 1993;O'Sullivan and Millero, 1998). Briefly, the AIRICA's high precision syringe pump draws a sample into a stripper that is pre-loaded with acid, effectively reducing the sample pH below 4.5 and converting all DIC to $CO_2$. A carrier gas strips the $CO_2$ from the acidified sample and the gas stream flows through a Peltier-element cooled condenser, followed by a Nafion dryer (both reducing water vapour content) before measurement by a NDIR analyser (LI-COR LI-7000, LI-COR Inc., Lincoln, NE, USA). DIC concentrations are determined by integrating the $CO_2$ mixing ratio signal in the NDIR analyser over the integration period, with the area versus concentration relationship established by 5-point calibration using in-house standards. The calibration factor was validated against Dickson certified reference materials (CRMs). The carrier gas used comprised of instrument air, with a background $CO_2$ mixing ratio of 289 ppm and a $\delta^{13}C$-$CO_2$ of -10.1 ‰. This carrier gas was used as the CRDS requires a gas with a similar composition to air. Furthermore, as the lower specified concentration range of the CRDS is ~ 380 ppm, the background $CO_2$ enables accurate measurement of a $\delta^{13}C$-$CO_2$ over a greater proportion of the integration period (discussed below).

The CRDS uses a laser-based spectroscopic technique that measures the individual carbon isotopologues ($^{12}C$ and $^{13}C$) of $CO_2$ at ~ 1 Hz and converts to standard $\delta^{13}C$ (‰) notation (referenced to Vienna PeeDee Belemnite). When the instrument is set in the $CO_2$ isotope only operating mode as used in this experiment, the guaranteed precision of the instrument by the manufacturer is 0.05 % for the $CO_2$ concentration (ppm) and 0.12 ‰ for $\delta^{13}C$ (1σ, 5 min average) spanning 380 to 2000 ppm. $\delta^{13}C$-DIC was determined using the $\delta^{13}C$ values of the extracted $CO_2$ measured during the integration period (see Determination of $\delta^{13}C$-DIC).

The coupling of the two instruments was achieved by simply linking the outlet of the AIRICA's NDIR analyser with the inlet of the CRDS using polyethylene lined Bev-A-Line® IV (1/8" ID, 1/4" OD) tubing (Fig. 1). A vent ensured no pressure build up at the inlet of the CRDS or backpressure on the AIRICA as the gas flow rate for the AIRICA varied from 70 to 300 ml min$^{-1}$ (discussed below), while the CRDS has a flow rate of ~ 35 ml min$^{-1}$. An additional drying agent (magnesium perchlorate) was used to dry the gas stream prior to entering the CRDS to minimise any potential artefacts in isotope values introduced by uncertainties in the manufacturers in-built water vapour corrections (Nara et al., 2012). Magnesium perchlorate was used as it does not induce a delay in $CO_2$ response time as is the case with some other desiccants (e.g. Drierite) (Webb et al., 2016).

To maximise precision and accuracy, DIC and $\delta^{13}C$-DIC were determined from separate injections. The measurement parameters were customised for each analysis such that the integration period for DIC and $\delta^{13}C$-DIC were 100 s and 310 s respectively. Therefore, in order to achieve autonomous measurement of DIC and $\delta^{13}C$-DIC, a computer script using AutoIt (Carvalho, 2016) was developed to autonomously modify the AIRICA's operating parameters at pre-determined time intervals without the need for an operator (the computer script is supplied as supplementary information). The script was sequenced so that a single measurement cycle consisted of the AIRICA system acidifying three aliquots of sample. The first aliquot was used to flush the system, the second aliquot was to determine DIC concentration, and the third aliquot was to determine $\delta^{13}C$-DIC. The system was flushed with carrier gas between each aliquot to remove the excess $CO_2$ and return to carrier gas concentrations. The cycle was then repeated, achieving a DIC and $\delta^{13}C$-DIC measurement, on





average, every ~ 16 minutes (refer to supplementary Figure S1 for a depiction of the typical CRDS output for a single measurement cycle).

### 2.3 Procedure

To determine DIC, the AIRICA's syringe pump was rinsed twice with 2100 µl of sample (filled and emptied at

300 µl s$^{-1}$) with the first rinse going directly to waste and the second rinse wasted via the stripper (Fig. 1). The syringe pump then drew 2000 µl of sample at 200 µl s$^{-1}$ which was then injected at 80 µl s$^{-1}$ into the stripper to which two drops of 10 % $H_3PO_4$ had been added. With a carrier gas flow rate through the stripper set a 300 ml min$^{-1}$, DIC concentrations were determined from integrating the $CO_2$ mixing ratio signal in the LICOR during an integration period of 100 s. The system was then flushed with carrier gas at 150 ml min$^{-1}$ to purge the

liberated $CO_2$ from the system and return to carrier gas values prior to $\delta^{13}$C-DIC sampling. Sampling volume for $\delta^{13}$C-DIC was the same as for DIC, however, to obtain a longer integration period the following AIRICA parameters were adjusted: the rate the sample was injected from the syringe pump to the stripper (i.e. injection rate) was reduced to 15 µl s$^{-1}$ (from 80 µl s$^{-1}$); the carrier gas flow rate through the stripper was reduced to 70 ml min$^{-1}$ (from 300 ml min$^{-1}$); and the integration period was increased to 310 s (from 100 s). $\delta^{13}$C-DIC was

determined from the $\delta^{13}$C-$CO_2$ data measured at ~ 1Hz during the integration period (discussed below). After the sampling for $\delta^{13}$C-DIC was completed, the cycle was restarted autonomously using the custom AutoIt script.

### 2.4 Determination of $\delta^{13}$C-DIC

The $\delta^{13}$C-$CO_2$ of the gas stream is a function of the carrier gas and that of the liberated $CO_2$ from the acidified sample (Eq. 1):

$$\delta^{13}C_{total} = (\delta^{13}C_{carrier} \times frac\ CO_{2carrier}) + (\delta^{13}C_{sample} \times frac\ CO_{2sample}) \tag{1}$$

whereby

$$frac\ CO_{2carrier} = CO_{2carrier} / CO_{2total}$$

$$frac\ CO_{2sample} = CO_{2sample} / CO_{2total}$$

$$CO_{2sample} = CO_{2total} - CO_{2carrier}$$

where $\delta^{13}C_{total}$ is the $\delta^{13}$C-$CO_2$ of the measured gas stream $CO_2$ (‰); $\delta^{13}C_{carrier}$ is the $\delta^{13}$C-$CO_2$ of the carrier gas $CO_2$ (‰); $CO_{2carrier}$ is the $^{12+13}CO_2$ concentration of the carrier gas (ppm); $\delta^{13}C_{sample}$ is the $\delta^{13}$C-$CO_2$ of the acidified sample (‰); $CO_{2sample}$ is the $^{12+13}CO_2$ concentration of the acidified sample (ppm); and $CO_{2total}$ is the $^{12+13}CO_2$ concentration of the measured gas stream (ppm).

The $\delta^{13}$C-CO2 values of $CO_2$ concentrations less than 400 ppm were excluded due to the guaranteed

specifications of the instrument spanning 380 to 2000 ppm. Of the remaining $\delta^{13}$C-$CO_2$ values, a mass balance was then used (to account for the $\delta^{13}$C value of the carrier gas) to determine a sample $\delta^{13}$C-$CO_2$ (i.e. $\delta^{13}C_{sample}$) value based on each measured $CO_2$ concentration (Eq. 2):

$$\delta^{13}C_{sample} = [\ (\delta^{13}C_{total} \times CO_{2total}) - (\delta^{13}C_{carrier} \times CO_{2carrier})\ ] / (CO_{2total} - CO_{2carrier}) \tag{2}$$



Five iterations of outlier removal were conducted on the ~ 1 Hz $\delta^{13}C_{sample}$ values, with removal of values with an absolute difference (versus the mean of all $\delta^{13}C_{sample}$ values for the sample) greater than two times the standard deviation of the sample. The remaining $\delta^{13}C_{sample}$ values were then averaged to determine the final $\delta^{13}C$-DIC value of the sample. All analysis was undertaken using Matlab (The Mathworks Inc., Natick, MA, USA) (see supplementary information for script).

### 2.5 Evaluation of precision and accuracy

$\delta^{13}C$-DIC standards were made using $Na_2CO_3$ for the isotopically heavy standard (-3.2 ± 0.1 ‰) and $K_2CO_3$ for the depleted standard (-26.8 ± 0.1 ‰) with both solids verified by IRMS using the international reference materials NBS-19 and LSVEC. The $Na_2CO_3$ solid was used to make a set of five standard solutions ranging from ~500 to ~3600 µmol kg$^{-1}$ and one standard solution was made using the $K_2CO_3$ solid (~2000 µmol kg$^{-1}$). Precision was evaluated based on the standard deviation of at least six replicate measurements for each standard. Accuracy was tested by comparing AIRICA-CRDS $\delta^{13}C$-DIC values to IRMS measured values from discrete samples collected during the bloom experiment (below). Both the precision measurements and bloom experiments were undertaken in a temperature-controlled laboratory with temperature ranging < 2 °C over the course of the experiments.

### 2.6 Algal bloom experiment

An algal bloom experiment was conducted to test the response of the method (Fig. 1) over an ~ 8 day period. Seawater (salinity 35.69 ppt) was collected from a nearby beach (28°49'22.01"S, 153°36'23.48"E) the morning the experiment commenced (19/01/2016). The water was enriched with 64 µmol l$^{-1}$ each of nitrate ($NO_3^-$) and orthosilicate ($SiO_4^{4-}$), and 4 µmol l$^{-1}$ of orthophosphate ($PO_4^{3-}$). In order to counteract the impact of the $SiO_4^{4-}$ addition on total alkalinity, 128 µmol l$^{-1}$ of hydrochloric acid (HCl) was added. The mesocosm was incubated under high pressure sodium bulbs (400W Phillips Son T Agro) at ~ 200 µmol m$^{-2}$ s$^{-1}$ for 18 hrs per day (12:00 am to 6:00 am).  The surface of the incubation vessel was covered with a transparent sheet of plastic film to inhibit evaporation and $CO_2$ exchange with the atmosphere allowing for an interpretation of the results in terms of a closed system. Macroalgae (*Ulva sp.*) and an additional dose of nutrients, were added to the sample water on the 25/1/16 (10:00 am) to further enhance biological activity. To achieve unattended sampling, incubated water was continuously recirculated through a sealed Schott bottle (250 ml) at ~ 1 l min$^{-1}$, from where the AIRICA's high precision syringe pump drew the sample (Fig. 1).  DIC and $\delta^{13}C$-DIC were sampled autonomously according to the procedure outlined above. In order to evaluate the accuracy of the AIRICA-CRDS method, 19 discrete samples for $\delta^{13}C$-DIC were collected throughout the bloom experiment for IRMS analysis. The samples were generally collected at the start and end of the light period from the return line of the recirculating system into 40 ml pre-combusted borosilicate vials, closed without headspace by Teflon-lined septa screw caps, poisoned with 50 µl of saturated $HgCl_2$ solution and stored at ~ 4 °C in the dark until analysis (within a week). The samples were also analysed for $\delta^{13}C$-DOC to determine the isotopic composition of the carbon pool available for microbial respiration. Samples were analysed using an OI Aurora 1030W TOC analyser interfaced to a Thermo Delta V$^{Plus}$ Isotope Ratio Mass Spectrometer (IRMS) (Maher and Eyre, 2011). Precision of $\delta^{13}C$-DIC and $\delta^{13}C$-DOC IRMS measurements were ± 0.1 ‰ and ± 0.3 ‰ respectively. Samples for DOC concentration were analysed using a Shimadzu (TOC-VCPH) analyser. Particulate organic carbon (POC,





mostly representing phytoplankton) was sampled at the end of the experiment by filtering 150 ml of the incubated water through pre-combusted GF/F filters and dried (60 °C) before analysis. Macroalgae was sampled at the end of the experiment, dried (60 °C) and ground prior to analysis. The $\delta^{13}C$ values of the macroalgae and the POC were measured in triplicate samples using a Thermo Flash EA coupled to a Delta V$^{plus}$ IRMS (± 0.1

‰). All $\delta^{13}C$ values are reported based on the VPDB scale.

### 3 Results and discussion

#### 3.1 Precision

The AIRICA-CRDS system exhibited a relatively small but detectable concentration effect on measured $\delta^{13}C$-DIC with an increase of 0.19 ‰ observed from 1000 µmol kg$^{-1}$ to 3600 µmol kg$^{-1}$ (Fig. 2A). It is unclear if this

effect is an artefact of the system or if it can be explained by ingassing of isotopically lighter atmospheric $CO_2$ (due to the high total alkalinity to DIC ratio of the standards) during the making of standards. $CO_2$ ingassing would have a larger effect on lower concentration standards (making them more depleted) than higher concentration standards, which follows the pattern observed here (Fig. 2A). Simarly, Bass et al. (2012) reported a concentration effect of < 0.2 ‰ increase from 1000 µmol kg$^{-1}$ to 2100 µmol kg$^{-1}$ and did not correct $\delta^{13}C$-DIC

values as the observed effect was within the precision of internal calibration standards.

The AIRICA-CRDS method had decreasing $\delta^{13}C$-DIC uncertainty with higher DIC concentrations (Fig. 2B). Standard deviations of the isotopically heavy standards decreased from ± 0.17 ‰ at ~ 1000 µmol kg$^{-1}$ (n=6) to ± 0.07 ‰ at ~ 3600 µmol kg$^{-1}$ (n=8), however, below 1000 µmol kg$^{-1}$ standard deviations were relatively high (± 0.63 ‰ at 500 µmol kg$^{-1}$, n=6). For the isotopically lighter standard, the uncertainty was ± 0.17 ‰ at a DIC

concentration of ~ 2000 µmol kg$^{-1}$ (Supplementary Table 1). Bass et al. (2012) also reported decreasing uncertainty with increasing concentrations of DIC. Their membrane based equilibration system attained standard deviations < ± 0.2 ‰ for concentrations above 360 µmol kg$^{-1}$ using a sample volume of 350 mL and an equilibration time of 720 s. In comparison, AIRICA-CRDS achieved < ± 0.2 ‰ uncertainty at ~ 1000 µmol kg$^{-1}$ on a sample volume of 2 mL and an integration time of 310 s.

To achieve the manufacturer guaranteed specifications of ± 0.12 ‰ $\delta^{13}C$-$CO_2$, $CO_2$ concentrations in the gas stream, which is a sum of the $CO_2$ in the carrier gas and the liberated $CO_2$ from the acidified sample, should be between 380 to 2000 ppm. The concentration of the liberated $CO_2$ is a function of the sample size, the injection rate, and the gas flow rate, each of which can be independently adjusted by the user through the AIRICA software. For $\delta^{13}C$-DIC measurements, the 2 ml of sample was injected into the stripper at 15 µl s$^{-1}$ and

extracted with a carrier gas flow rate of 70 ml min$^{-1}$ (achieving a total measurement rime of 310 s), resulting in the AIRICA-CRDS achieving an average precision of 0.14 ± 0.04 ‰ (n = 84) for all standards above 1000 µmol kg$^{-1}$. While we optimised the system for coastal and oceanic DIC concentrations, if sampling low DIC concentrations (i.e. < 1000 µmol kg$^{-1}$), similarly precise $\delta^{13}C$-DIC values may be achievable if, for example, a larger syringe volume is used to increase sample size. This ability for customisation adds to the functionality of

the AIRICA-CRDS system. In comparison to the AIRICA-CRDS reported here, a worldwide proficiency test of $\delta^{13}C$-DIC analysis found laboratory precision ranged from 0.1 to 0.5 ‰ depending on different methodologies





(Van Geldern et al., 2013). However, the inter-laboratory results revealed average standard deviations of $\pm$ 0.45 ‰ and $\pm$ 0.47 ‰ for $\delta^{13}$C-DIC values for lake water and seawater measurements respectively.

### 3.2 Accuracy

Accuracy of the AIRICA-CRDS $\delta^{13}$C-DIC was determined by direct comparison with IRMS measurements collected simultaneously during the algal bloom experiment (Fig. 3). The average difference in measured $\delta^{13}$C values between methods was 0.1 $\pm$ 0.06 ‰ which is similar to the accuracy of the IRMS (0.1 ‰). This robust relationship was based on DIC concentrations ranging from 1986 $\mu$mol kg$^{-1}$ to 2232 $\mu$mol kg$^{-1}$ (average 2129 $\mu$mol kg$^{-1}$) during the dynamic bloom experiment. This demonstrates that the automated AIRICA-CRDS system described here attains similar accuracy to IRMS $\delta^{13}$C-DIC measurements at typical coastal and oceanic DIC concentrations.

### 3.3 Bloom experiment

The performance of the AIRICA-CRDS to characterise changes in $\delta^{13}$C-DIC values and DIC concentrations in the marine environment is demonstrated by the algal bloom experiment (Fig. 4A). A total of 664 DIC concentrations and 661 $\delta^{13}$C-DIC values were autonomously measured during the ~ 8 day incubation. Concentrations of DIC ranged from 1965 to 2253 $\mu$mol kg$^{-1}$ and $\delta^{13}$C-DIC ranged from 0.96 ‰ to -3.61 ‰.

The AIRICA-CRDS method captured a temporally inverse relationship between $\delta^{13}$C-DIC values and DIC concentrations throughout the algal bloom experiment. During the first 3 days respiration was the dominant metabolic process releasing isotopically lighter DIC (originating from organic matter). Respiration increased the pool of DIC from ~ 1995 to ~ 2230 $\mu$mol kg$^{-1}$ and simultaneously depleted $\delta^{13}$C-DIC from ~ 0.7 ‰ to ~ -2.7 ‰. After 3 days diel cycling (light-dark) of DIC and $\delta^{13}$C-DIC commenced, likely due to the time lag associated with primary producer biomass increase after the initial addition of nutrients. During daylight hours, photosynthetic production preferentially consumed the isotopically lighter $^{12}$CO$_2$ component of the DIC pool, decreasing DIC concentrations and enriching $\delta^{13}$C-DIC values. Conversely, during the dark hours, respiration increased DIC concentrations and decreased $\delta^{13}$C-DIC values. Over the course of days 4, 5 and 6, overall net production drew down the DIC pool. On day 7 no dark incubation occurred resulting in a large photosynthetically-driven decrease in the DIC concentration from 2164 $\mu$mol kg$^{-1}$ to 1965 $\mu$mol kg$^{-1}$ and an enrichment of $\delta^{13}$C-DIC from ~ -3.1 ‰ to ~ -1.7 ‰.

### 3.4 Insights from high resolution DIC and $\delta^{13}$C-DIC measurements

The AIRICA-CRDS's high resolution measurements of $\delta^{13}$C-DIC can provide insights into drivers of DIC in aquatic environments. To illustrate a potential application of the AIRICA-CRDS approach, Keeling plots are used to interpret carbon sources during a simulated algal bloom experiment. Keeling plots (Keeling, 1958) visualize a simple two component mixing model and are commonly used to interpret sources of added carbon in aquatic, terrestrial and atmospheric sciences. Using a model II regression, the y-intercept of the regression between the inverse of DIC concentration and $\delta^{13}$C-DIC estimates the isotopic composition of the carbon source. The approach has been used to determine the isotopic composition of the DIC source in various settings including freshwater lakes (Karlsson et al., 2007), coral reefs (Carvalho et al., 2015), mangroves (Maher et al., 2013a) and groundwaters (Porowska, 2015).





The $\delta^{13}$C-DIC source/sink value were estimated by Keeling plots for each linear increase (respiration) and decrease (production) in DIC concentrations during the simulated algal bloom based on 5-point averages of measured DIC and $\delta^{13}$C-DIC (Fig. S2). The uncertainty in the intercept (i.e. the $\delta^{13}$C-DIC source value) decreased as the absolute change in DIC increased, achieving < 2 ‰ uncertainty when the change in DIC was

greater than ~ 40 µmol kg$^{-1}$ (Fig. 5). Keeling plots based on DIC concentrations during the first three incubation days, when respiration was the dominant process (Table 1, Fig. S2a,b,c), produced $\delta^{13}$C-DIC intercept values similar to that of terrestrial C3 vegetation (Smith and Epstein, 1971) (-25.9 ± 1.8 ‰, -29.3 ± 0.4 ‰ and -33.3 ± 0.6 ‰). In contrast, the $\delta^{13}$C-DIC intercept value for the final dark incubation of -17.4 ± 0.9 ‰ (Table 1, Fig. S2f) is similar to that of marine organic matter (OM). Furthermore, $\delta^{13}$C-DIC intercept values determined from

linear decreases in DIC (i.e. photosynthetic carbon fixation) are also similar to marine OM (Fry and Sherr, 1989) (-17.4 ± 2 ‰ and -19.0 ± 0.3 ‰, Table 1, Fig. S2d,e). Thus, results suggest a distinct shift in the source of DIC during the course of the experiment, from terrestrially derived OM to marine derived OM. This is likely due to a shift towards a dominance of marine organic matter toward the end of the experiment as a result of the algal bloom and the added macroalgae (added on day 6, Fig. 4).

The DOC results support our interpretation of a shift in the DIC source. DOC concentrations increased from 88 µmol kg$^{-1}$ at the beginning of the experiment to 364 µmol kg$^{-1}$ at the end. $\delta^{13}$C-DOC values became more depleted for the first 3 days of the experiment before becoming more enriched during the final ~ 2 days. Keeling plot intercepts based on DOC concentrations for the initial ~3 day period (-31.8 ± 1.5 ‰) and final ~ 2 day period (-17.6 ± 2 ‰) suggests a distinct shift in the carbon source contributing to the DOC pool from a

terrestrially derived DOC source to a marine source. This is consistent with the shift observed in the DIC pool. The IRMS $\delta^{13}$C-POC results for phytoplankton (-25.7 ‰) and macroalgae (i.e. ulva, -10.1 ‰) suggests a similar contribution of both sources to the final $\delta^{13}$C-DOC intercept value of -17.6 ‰ and the $\delta^{13}$C-DIC value of -17.38 ‰. The AIRICA-CRDS system could similarly be used to characterise the importance of various other processes, such as, the relative importance of organic matter respiration versus carbonate dissolution as a source

of DIC from coastal systems (Carvalho et al., 2015) and the importance of allocthonous versus autochthonous organic matter for supporting bacterial productivity (Guillemette et al., 2013).

## 4 Comments

The utility of the AIRICA-CRDS method for autonomous, high resolution measurements of DIC and $\delta^{13}$C-DIC in a laboratory configuration has been demonstrated. Yet, the simple system also has the capacity for field

deployment. Both of the commercially available instruments (coupled without any modifications to their hardware) have been used in field-based studies. CRDS have been deployed in a range of environmental settings including large research vessels (Bass et al., 2014b;Becker et al., 2012), vehicles (Maher et al., 2014), and small boats (Maher et al., 2015). Determination of DIC based on the NDIR gas analysis method has been used to conduct continuous shipboard measurements (Friederich et al., 2002;Hiscock and Millero, 2005) and the

AIRICA system has been deployed on research vessels to determine concentrations of DIC from discrete samples (Balch et al., 2016;Bates et al., 2014;Bates et al., 2013). Assuming access to an appropriate power source and suitable environmental conditions to house the instruments, the AIRICA-CRDS system has potential to be deployed in the field.



The AIRICA-CRDS is an alternative system that enables autonomous, high resolution measurements of DIC and $\delta^{13}$C-DIC with precision and accuracy comparable to traditional IRMS techniques. In the described configuration, the system achieved a sampling resolution of 16 mins, with a DIC precision of $\pm$ 1.5-2 µmol kg$^{-1}$ and $\delta^{13}$C-DIC precision of $\pm$ 0.14 ‰ for concentrations spanning 1000 to 3600 µmol kg$^{-1}$. A change in DIC concentrations of ~ 40 µmol kg$^{-1}$ was sufficient to obtain a precision of < 2 ‰ in source interpretations based on Keeling plots. Whilst we optimized the system for typical estuarine/marine DIC concentrations, the system has the potential to be optimised for other environments through changes to the syringe size, sample volume and injection speed. This ability for customisation adds to the functionality of the AIRICA-CRDS system, offering the potential to explore complex carbonate systems question across a range of aquatic settings.

**Author contribution**

D. T. Maher, K. G. Schulz and I. R. Santos designed the study. D. T. Maher, K. G. Schulz and M. Call conducted the experiments. M. C. Carvalho developed the AutoIT script and analysed all discrete samples. K. G. Schulz developed the Matlab Scripts. M. Call prepared the manuscript with contributions from all authors.

The authors declare that they have no conflict of interest.

**Acknowledgements**

We acknowledge funding from the Australian Research Council through grants DE140101733, DE150100581, LE120100156. Kai G. Schulz is the recipient of an Australian Research Council Future Fellowship (FT120100384).

**Supporting information available**

Detailed supporting information is available. The supporting information includes the AutoIt script, Matlab script and supplementary figures and tables.

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



**Tables**

**Table 1. Keeling plot intercept values and standard deviations for the linear increase/decrease in DIC concentrations during the simulated algal bloom as indicated by the lower case letters in Figure 4. Intercept values are derived from the regression of 1/DIC concentrations vs. $\delta^{13}$C-DIC values based on 5-**

5  **point averages of measured DIC concentrations and $\delta^{13}$C-DIC values (see Fig. S2).**

|   | y-Intercept value (‰) |
|---|---|
| a | -25.92 ± 1.79 |
| b | -29.34 ± 0.43 |
| c | -33.29 ± 0.65 |
| d | -17.43 ± 1.96 |
| e | -19.05 ± 0.35 |
| f | -17.38 ± 0.93 |




**Figures**

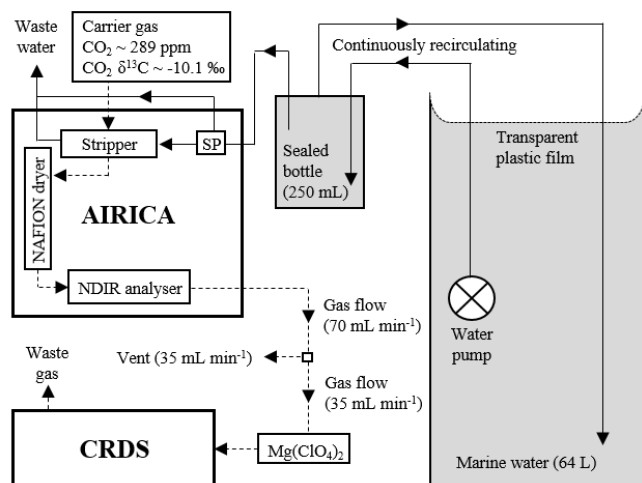

**Figure 1. Schematic of the coupled Autonomous Infra Red Inorganic Carbon Analyser (AIRICA) and**
5      **Cavity Ring Down Spectrometer (CRDS) set up to autonomously and continuously measure DIC**
**concentrations and δ$^{13}$C-DIC values. Solid arrows indicate liquid flow and dashed arrows indicate gas**
**flow. SP = syringe pump.**

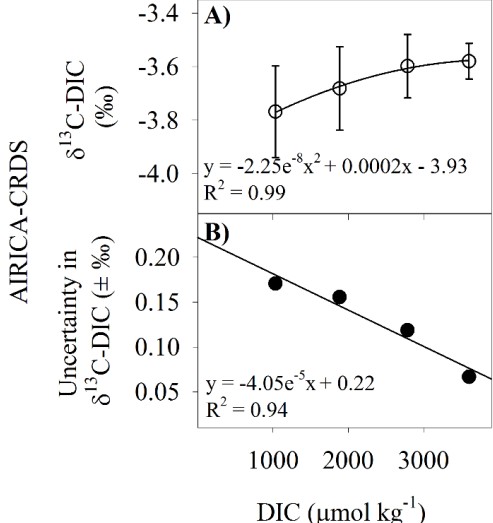

10     **Figure 2. A) Concentration effect on δ$^{13}$C-DIC data produced by the AIRICA-CRDS B) Uncertainty in**
**δ$^{13}$C-DIC versus DIC concentration.**





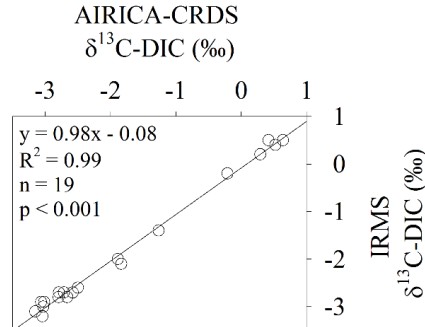

**Figure 3. Comparison of ARICA-CRDS δ¹³C-DIC vs IRMS δ¹³C-DIC.**





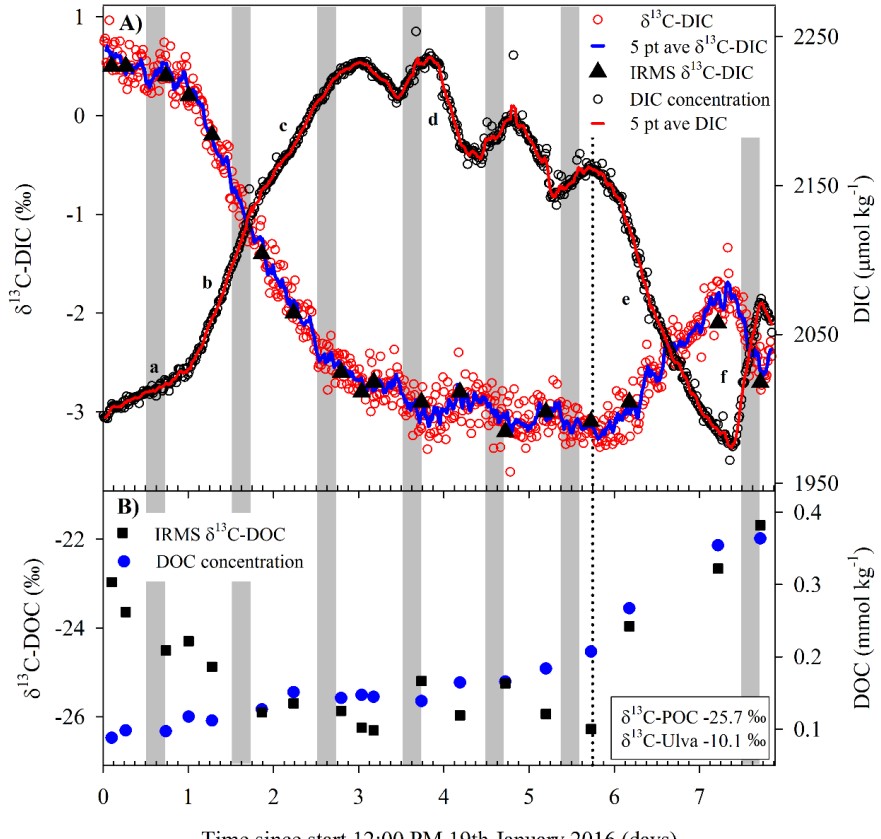

**Figure 4. A)** δ¹³C-DIC values and DIC concentrations measured by the AIRICA-CRDS system during the
~ 8 day laboratory bloom experiment. Blue and red lines are 5 point averages for δ¹³C-DIC and DIC
respectively. Black triangles are discrete IRMS δ¹³C-DIC values plotted for comparison. Shaded areas
indicate dark incubations. Lower case letters indicate the sections used for Keeling plots (see Fig. S2).
Dotted line is when macroalgae (*Ulva sp.*) and additional nutrients were added. **B)** IRMS δ¹³C-DOC
values and DOC concentrations for discrete samples. Inset box displays IRMS δ¹³C values for POC and
*Ulva sp*.



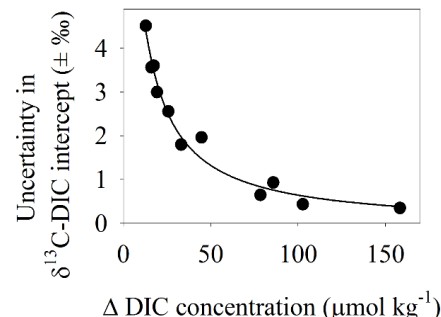

**Figure 5. Relationship between Keeling intercept uncertainty and changes in DIC concentration. Higher changes in DIC resulted in lower uncertainties in Keeling plot interpretations.**