# Peer review of "TECHNICAL NOTE: Coupling infrared gas analysis and cavity ring down spectroscopy for autonomous, high temporal resolution measurements of DIC and $\delta^{13}$ C-DIC"

_Biogeosciences, 2016_

## Referee Comment (RC1) · Anonymous Referee #1 · 7 Dec 2016

Review of Call et al. Coupling infrared gas analysis ..

The brief technical note by Call et al. presents an interesting case of coupling two commercially available instruments for combined analysis of [DIC] and $\delta$13C-DIC analysis that may have a wide range of applications under both lab and field conditions. This team has been very active in these novel applications and the technical aspects of the study appear to be very sound and the intercalibration with IRMS measurements is promising. I also welcome the fact that the authors made ample technical information and scripts available for those who wish to implement this approach. My main concern is their interpretation of the data resulting from the 8-day incubation experiments which is used as an example application. The Keeling approach used is in my opinion not appropriate to apply to this dataset since it is only applicable to period where only respiration occurs: then you have a situation where a 2-source mixing model applies. During periods of illumination, when primary production is important, this principle does not hold and applying a Keeling approach is not valid, what is happening here is a case of isotope fractionation, not isotope mixing.

The authors should thus only apply the Keeling approach to 'nighttime' data. Also, more information should be added on how the Keeling method was applied, i.e. what type of regressions were used. There is a wealth of literature on the importance of using a correct regression method, Pataki et al. (2003) and Zobitz et al (2006) are a good start:

Zobitz JM, Keener JP, Schnyder H, & Bowling DR (2006) Sensitivity analysis and quantification of uncertainty for isotopic mixing relationships in carbon cycle research. Agricultural and Forest Meteorology 136: 56-75

Pataki, D.E., Ehleringer, J.R., Flanagan, L.B.D.Y., Bowling, D.R., Still, C.J., Buchmann, N., Kaplan, J.O., Berry, J., 2003b. The application and interpretation of Keeling plots in terrestrial carbon cycle research. Global Biogeochem. Cycles 17 (1), 1022, doi:10.1029/2001GB001850

Minor suggestions:

-p3: why is 'instrument air' needed or used, and not a $CO_2$-free carrier gas ? Would this not simplify and improve measurements, or is there a reason I'm overlooking that a certain background level of $CO_2$ is required ? If not, you could simply strip out the $CO_2$ with either a cold trap or a $CO_2$ scrubber.

-p5: explain in more detail how the standards were prepared – preparing these requires some precautions in terms of removing all dissolved $CO_2$ prior to dissolving

your powdered standards etc.

-p5: salinity has no units, remove 'ppt'

-p7 and further throughout the ms: use correct terminology when referring to higher or lower d13C values, e.g. L19: 'depleted d13C-DIC' should be 'lowered d13C-DIC', L23: 'enriching d13C-DIC values' should be 'increasing d13C-DIC values', L27: 'enrichment of d13C-DIC' should be 'increase of d13C-DIC' etc.

-p8: see initial commens on Keeling plot approach: (i) provide details on regression techniques, and (ii) should not be applied on data from periods with primary production.

-P8 L21: ulva → Ulva sp. (capital, italics)

P8 L22 : use one decimal only for d13C data

---

## Referee Comment (RC2) · N. Munksgaard (Referee) · 16 Dec 2016

N. Munksgaard (Referee)

niels.munksgaard@jcu.edu.au

General comments:

This article describes the analytical procedure and performance of two commercial instruments coupled to provide near-simultaneous, high quality data on DIC concentrations and DIC 12C/13C isotope ratios (delta 13C). Similar automated and high-frequency data have previously been obtained using custom designed instruments in both the laboratory and the field (as referenced). The merit of the approach described

here is that the interfacing of the two instruments, each optimised for their respective analysis, provides combined data with improved precision and accuracy compared to the previously described methods.

The authors convincingly demonstrate that the technique works well in the laboratory and comment that it may also be field deployable with adequate power and shelter. However, the practicalities of field deployment would not be easy as the several components (CRDS, AIRICA, LICOR, PC, monitor, air tank, power etc) must weigh 50-60 kg. Also, the authors describe their system as 'simple' which might be slightly 'optimistic', especially in a field setting. Further testing would be needed to demonstrate performance in the field (e.g. sensitivity to temperature changes).

I find this a high quality technical note that reads very well and the procedures are mostly clearly described, it is also great that the supplement includes computer coding to aid other researchers setting up this technique.

I'm curious as to why 'zero air' ($CO_2$ free air) was not used as carrier – instead, air with a reduced $CO_2$ content was used requiring a mass balance calculation to derive the sample isotope results. I would think this potentially degrades performance.

Regarding the previous comment (ref #1) on the use of the Keeling plot for the linear sections of Fig. 4 (discussion P 8 re respiration and photosynthetic fixation during the algae bloom experiment), I agree that this treatment seem valid for the dark sections (respiration) – here the two mixing components would be (1) the existing DIC pool and (2) the added DIC (respired $CO_2$ – although this could be a constant mixture of $CO_2$ coming from more than one source). For the light sections (photosynthesis), the 'mixing' line is in effect an 'un-mixing' line ($CO_2$ - and preferentially $^{12}CO_2$ – being removed from the DIC pool). How such a line should be interpreted seems highly uncertain given the associated (and uncertain/variable) isotopic fractionation effects. I suggest modifying this section of the discussion but an exhaustive explanation should not be necessary in this technical note.

Specific comments: 1. P3 line 12-14: need more specifics for the 'Dickson CRM', justify why CO2 free air was not used 2. P3 line 17: 'CO2 only operating mode' is confusing here – is it because this instrument also can measure CH4? 3. P5: May point out that blooming algae would have been present in the sampled seawater – is there any information on the type/species? 4. P7 line 21: Similar systematic changes in DIC and d13C was previously described for coral by the Bass et al 2012 study 5. P8: For the algae bloom experiment, additional details are required on how the uncertainty of the intercept d13C values in the Keeling plots were derived (Fig. S2) – the uncertainties should be added to the figures. 6. P8 line 3: It should be emphasised that uncertainty of the intercept d13C value is very dependent on the range of [DIC] in each plot – and much higher than the uncertainty of the individual d13C DIC data points. Could this uncertainty be improved by manipulating carrier flow and sample size etc to increase the [DIC] range? Would using CO2 free air as carrier increase the range? 7. P8 line 10: Suggest expanding this explanation a bit: initial source is terrestrial OM present in the sampled coastal seawater, then marine OM from the Ulva sp introduced later in the experiment 8. Figure 3 shows a very good correlation between AIRICA-CRDS and IRMS results for samples, yet supplement Table S1 seems to show an offset of 0.3-0.5 ‰ between the two techniques 9. Supplement Fig. S1: Seems surprising that rinse, DIC and d13C cycles produce same concentration (peak value) - is the concentration limited by settings of the AIRICA?

---

## Author Comment (AC1) · 8 Feb 2017

We thank both referees for their positive and constructive reviews which have helped improve our manuscript. We have revised the manuscript with the changes outlined in the following.

**General comments by Anonymous Referee # 1**

The brief technical note by Call et al. presents an interesting case of coupling two commercially available instruments for combined analysis of [DIC] and δ13C-DIC analysis that may have a wide range of applications under both lab and field conditions. This team has been very active in these novel applications and the technical aspects of the study appear to be very sound and the intercalibration with IRMS measurements is promising. I also welcome the fact that the authors made ample technical information and scripts available for those who wish to implement this approach. My main concern is their interpretation of the data resulting from the 8-day incubation experiments which is used as an example application. The Keeling approach used is in my opinion not appropriate to apply to this dataset since it is only applicable to period where only respiration occurs: then you have a situation where a 2-source mixing model applies. During periods of illumination, when primary production is important, this principle does not hold and applying a Keeling approach is not valid, what is happening here is a case of isotope fractionation, not isotope mixing.

The authors should thus only apply the Keeling approach to 'nighttime' data. Also, more information should be added on how the Keeling method was applied, i.e. what type of regressions were used. There is a wealth of literature on the importance of using a correct regression method, Pataki et al. (2003) and Zobitz et al (2006) are a good start:

Zobitz JM, Keener JP, Schnyder H, & Bowling DR (2006) Sensitivity analysis and quantification of uncertainty for isotopic mixing relationships in carbon cycle research. Agricultural and Forest Meteorology 136: 56-75

Pataki, D.E., Ehleringer, J.R., Flanagan, L.B.D.Y., Bowling, D.R., Still, C.J., Buchmann, N., Kaplan, J.O., Berry, J., 2003b. The application and interpretation of Keeling plots in terrestrial carbon cycle research. Global Biogeochem. Cycles 17 (1), 1022, doi:10.1029/2001GB001850

Whilst Keeling plots have been applied to periods when $CO_2$ sources (respiration) and sinks (primary production) occur simultaneously, that is, during periods of illumination (Vardag et

al., 2016), we believe a discussion on the validity of this approach is beyond the scope of this technical note. Thus, we have amended the manuscript such that the Keeling approach was applied only to 'nighttime' periods, i.e. when only respiration occurs and the DIC isotope signal is purely a function of mixing.

With regards to the type of regression selected, the original manuscript did state that that a model II regression was used (L33, P7). Again, we believe that detailing the specific aspects of the regression models available is beyond the scope of this technical note, however, we have now referenced the review by Pataki et al. (2003) and provided additional detail:

*"Keeling plots (Keeling, 1958) visualize a simple two component mixing model and are commonly used to interpret sources of added carbon in aquatic, terrestrial and atmospheric sciences (see (Pataki et al., 2003) for details on underlying assumptions and types of regression models available). Using a Model II regression (which assumes errors in the measurement of both variables), the y-intercept of the regression between the inverse of DIC concentration (1/DIC) and $\delta^{13}C$-DIC estimates the isotopic composition of the carbon source. The approach has been used to determine the isotopic composition of the DIC source in various settings including freshwater lakes (Karlsson et al., 2007), coral reefs (Carvalho et al. 2015), mangroves (Maher et al., 2013a) and groundwaters (Porowska, 2015)."*

Minor suggestions from Anonymous Reviewer # 1:

-3: why is 'instrument air' needed or used, and not a CO2-free carrier gas? Would this not simplify and improve measurements, or is there a reason I'm overlooking that a certain background level of CO2 is required? If not, you could simply strip out the CO2 with either a cold trap or a CO2 scrubber.

Briefly, to achieve the precision and accuracy obtained by the AIRICA-CRDS based on the $CO_2$ extracted from a small sample (2 ml), we used a gas stream with a background level of $CO_2$. This allowed for a greater proportion of the integration period to be within the manufacturer specifications (380 ppm is the minimum concentration for the guaranteed precision of the CRDS instrument by the manufacturer). Thus, a mass balance was required to account for the $\delta^{13}C$ value of the carrier gas to determine a $\delta^{13}C$-DIC. This is now explained in greater detail throughout the manuscript.

The decision to use instrument air (as opposed to $CO_2$-free carrier gas) was first outlined at P3 L10-15: *"The carrier gas used comprised of instrument air, with a background $CO_2$ mixing ratio of 289 ppm and a $\delta^{13}C$-$CO_2$ of -10.1 ‰. This carrier gas was used as the CRDS requires a gas with a similar composition to air. Furthermore, as the lower specified concentration range of the CRDS is ~ 380 ppm, the background $CO_2$ enables accurate measurement of a $\delta^{13}C$-$CO_2$ over a greater proportion of the integration period…"*

We acknowledge that both Reviewer's had similar comments as to why instrument air was used. Therefore, we have provided additional detail in the aforementioned text to clarify the use of a $CO_2$-free carrier gas earlier in the manuscript:

*"The carrier gas used comprised of instrument air (as opposed to $CO_2$-free carrier gas), with a background $CO_2$ mixing ratio of 289 ppm and a $\delta^{13}C$-$CO_2$ of -10.1 ‰. This carrier gas was used as the CRDS requires a gas with a similar composition to air. Furthermore, as the lower specified concentration range of the CRDS is ~ 380 ppm, to achieve the precision and*

*accuracy obtained by the AIRICA-CRDS from the $CO_2$ extracted from a small sample (2 ml), the background $CO_2$ enables accurate measurement of a $\delta^{13}C\text{-}CO_2$ over a greater proportion of the integration period (discussed below)."*

-p5: explain in more detail how the standards were prepared – preparing these requires some precautions in terms of removing all dissolved CO2 prior to dissolving your powdered standards etc.

Standards were made by dissolving carbonate solids in ultra-pure water (18.2 Milli-Q®) which had a DIC concentration < 2 µmol kg$^{-1}$. We have added this sentence:

*"All standards were made by dissolving carbonate solids in ultra-pure water (18.2 Milli-Q®)."*

-p5: salinity has no units, remove 'ppt'

We have removed 'ppt' in the revised manuscript.

-p7 and further throughout the ms: use correct terminology when referring to higher or lower d13C values, e.g. L19: 'depleted d13C-DIC' should be 'lowered d13C-DIC', L23: 'enriching d13C-DIC values' should be 'increasing d13C-DIC values', L27: 'enrichment of d13C-DIC' should be 'increase of d13C-DIC' etc.

We have made these changes throughout the manuscript.

-p8: see initial comments on Keeling plot approach: (i) provide details on regression techniques, and (ii) should not be applied on data from periods with primary production.

Please refer to our aforementioned response to the initial comments.

-p8 L21: ulva ! Ulva sp. (capital, italics)

We have made this change in the revised manuscript.

-p8 L22 : use one decimal only for d13C data

We have made this change in the revised manuscript.

References used in this reply:

Martz, T. R., Daly, K. L., Byrne, R. H., Stillman, J. H., and Turk, D.: Technology for ocean acidification research: Needs and availability, Oceanography, 28, 40-47, 10.5670/oceanog.2015.30, 2015.
Pataki, D. E., Ehleringer, J. R., Flanagan, L. B., Yakir, D., Bowling, D. R., Still, C. J., Buchmann, N., Kaplan, J. O., and Berry, J. A.: The application and interpretation of Keeling plots in terrestrial carbon cycle research, Global Biogeochemical Cycles, 17, n/a-n/a, 10.1029/2001GB001850, 2003.
Vardag, S. N., Hammer, S., and Levin, I.: Evaluation of 4 years of continuous δ13C(CO2) data using a moving Keeling plot method, Biogeosciences, 13, 4237-4251, 10.5194/bg-13-4237-2016, 2016.

**General comments by Referee # 2, Neils Munksgaard**

This article describes the analytical procedure and performance of two commercial instruments coupled to provide near-simultaneous, high quality data on DIC concentrations and DIC 12C/13C isotope ratios (delta 13C). Similar automated and high-frequency data have previously been obtained using custom designed instruments in both the laboratory and the field (as referenced). The merit of the approach described here is that the interfacing of

the two instruments, each optimised for their respective analysis, provides combined data with improved precision and accuracy compared to the previously described methods.

The authors convincingly demonstrate that the technique works well in the laboratory and comment that it may also be field deployable with adequate power and shelter. However, the practicalities of field deployment would not be easy as the several components (CRDS, AIRICA, LICOR, PC, monitor, air tank, power etc) must weigh 50-60 kg. Also, the authors describe their system as 'simple' which might be slightly 'optimistic', especially in a field setting. Further testing would be needed to demonstrate performance in the field (e.g. sensitivity to temperature changes).

In reviewing the technologies available to ocean acidification scientists, Martz et al. (2015) noted that the replication of a described customised system can be "either untenably time consuming or completely intractable". Our description of the simplicity of the AIRICA-CRDS refers to the fact that our new method does not require the need to design or replicate a customised system. We believe that coupling two high precision, commercially available instruments without any modification to their hardware does represent simplicity.

Our comments regarding the potential for field deployment is based on both instruments being used in field settings previously. We acknowledge that further testing would need to demonstrate its performance in the field and as such, have added:

*"however, further testing is required to test this capability."*

I find this a high quality technical note that reads very well and the procedures are mostly clearly described, it is also great that the supplement includes computer coding to aid other researchers setting up this technique.

I'm curious as to why 'zero air' ($CO_2$ free air) was not used as carrier – instead, air with a reduced $CO_2$ content was used requiring a mass balance calculation to derive the sample isotope results. I would think this potentially degrades performance.

Please refer to the response provided to Anonymous Referee #1's similar comment.

Regarding the previous comment (ref #1) on the use of the Keeling plot for the linear sections of Fig. 4 (discussion P 8 re respiration and photosynthetic fixation during the algae bloom experiment), I agree that this treatment seem valid for the dark sections (respiration) – here the two mixing components would be (1) the existing DIC pool and (2) the added DIC (respired $CO_2$ – although this could be a constant mixture of $CO_2$ coming from more than one source). For the light sections (photosynthesis), the 'mixing' line is in effect an 'un-mixing' line ($CO_2$ - and preferentially $^{12}CO_2$ – being removed from the DIC pool). How such a line should be interpreted seems highly uncertain given the associated (and uncertain/variable) isotopic fractionation effects. I suggest modifying this section of the discussion but an exhaustive explanation should not be necessary in this technical note.

Please refer to our response to Anonymous Referee #1.

Specific comments:

1. P3 line 12-14: need more specifics for the 'Dickson CRM', justify why $CO_2$ free air was not used

We have added specifics for the Dickson CRM to that sentence:

*"(Batch # 136, DIC 2021.15 µmol kg$^{-1}$, Sal 33.678)"*

2. P3 line 17: 'CO2 only operating mode' is confusing here – is it because this instrument also can measure CH4?

Yes, the CRDS (Picarro G2201-*i*) can also simultaneously measure the carbon stable isotope ratio of methane ($\delta^{13}$C-CH$_4$). We have added the following to the sentence:

*"(the CRDS can also simultaneously determine the carbon stable isotope ratio of methane)"*.

3. P5: May point out that blooming algae would have been present in the sampled seawater – is there any information on the type/species?

No, species type was not determined. This level of detail is beyond the scope of the manuscript.

4. P7 line 21: Similar systematic changes in DIC and d13C was previously described for coral by the Bass et al 2012 study.

We have included the following reference to this paper in the revised manuscript:

*"Similar diel variations were observed for DIC and $\delta^{13}$C-DIC in a mesocosm coral reef metabolism experiment (Bass et al., 2012)."*

5. P8: For the algae bloom experiment, additional details are required on how the uncertainty of the intercept d13C values in the Keeling plots were derived (Fig. S2) – the uncertainties should be added to the figures.

We have updated Fig. S2 to include the uncertainties of both the slope and the intercept of the regression equation. Intercept uncertainty is based on the statistical uncertainty associated with the regression. The Model II regression (geometric mean regression) assumes significant errors in the measurement of both variables. Pataki et al. (2003) and references therein provides an explanation on the how the uncertainties are derived for this regression model and we have added this reference when discussing the intercept uncertainty.

6. P8 line 3: It should be emphasised that uncertainty of the intercept d13C value is very dependent on the range of [DIC] in each plot – and much higher than the uncertainty of the individual d13C DIC data points. Could this uncertainty be improved by manipulating carrier flow and sample size etc to increase the [DIC] range? Would using CO2 free air as carrier increase the range?

The uncertainty of the intercept value is very dependent on the rate of change in DIC concentration during the incubation period i.e. the magnitude of the respiration signal. We have changed the manuscript to emphasise this:

*"The uncertainty in the intercept (i.e. the $\delta^{13}$C-DIC source value) is dependent on the rate of change in DIC concentration during the dark incubation period. Uncertainty decreased as the absolute change in DIC increased, achieving < 2 ‰ uncertainty when the change in DIC was greater than ~ 40 µmol kg$^{-1}$ (Fig. 5)."*

We have also noted that the uncertainty of the intercept value is much higher than the uncertainty for each $\delta^{13}$C-DIC and DIC data value used to determine the intercept value:

*"It is noted that the intercept uncertainty is much higher than the uncertainty for each $\delta^{13}$C-DIC and DIC value ($\pm$ 0.14 ‰ and $\pm$ 1.5 to 2 µmol kg$^{-1}$ respectively)."*

The rate of change in DIC concentration is a function of the experimental conditions (in this case the respiration rate), thus, manipulating the individual features of the ARICA-CRDS (e.g. carrier flow, sample size, etc.) would not improve the intercept uncertainty.

7. P8 line10: Suggest expanding this explanation a bit: initial source is terrestrial OM present in the sampled coastal seawater, then marine OM from the Ulva sp introduced later in the experiment

We have amended a later sentence to:

*"Thus, results suggest a distinct shift in the source of DIC during the course of the experiment, from terrestrially derived OM present in the sampled coastal seawater to marine derived OM at the end. This is likely due to a shift towards a dominance of marine organic matter toward the end of the experiment as a result of the algal bloom and the added macroalgae (added on day 6, Fig. 4)."*

8. Figure 3 shows a very good correlation between AIRICA-CRDS and IRMS results for samples, yet supplement Table S1 seems to show an offset of 0.3-0.5 ‰ between the two techniques

Accuracy of the AIRICA-CRDS $\delta^{13}$C-DIC was determined by direct comparison with IRMS measurements collected simultaneously during the algal bloom experiment. Figure 3 shows these results.

Table S1 on the other hand shows results of the precision results. The table supplements the observed decreasing $\delta^{13}$C-DIC uncertainty with higher DIC concentrations is shown in Fig. 2B. The purpose of this Table was to highlight the precision based on repeated measurements of standards made at differing concentrations, rather than a determination of accuracy.

9. Supplement Fig. S1: Seems surprising that rinse, DIC and d13C cycles produce same concentration (peak value) - is the concentration limited by settings of the AIRICA?

Pre-testing allowed careful adjustment of gas flow rates and sample injection rates to ensure concentrations peaks were similar for both analysis types, with peak concentrations falling in the optimal range for both the AIRICA and CRDS instruments.

---

## Author Response (AR2)

We thank the Associate Editor, Dr Jack Middleburg, for his positive feedback and reviews which have helped improve our manuscript. We have revised the manuscript with the minor corrections:

p.1, abstract. Accuracy is reported in permille, indicate that this refers to del13C to avoid confusion with accuracy of DIC.

We added "for δ13C-DIC" so it now reads:

*"Accuracy of 0.1 ± 0.06 ‰ for $\delta^{13}C$-DIC based on DIC concentrations ranging from 2000 µmol kg-1 to 2230 µmol kg$^{-1}$ was achieved during a laboratory-based algal bloom experiment."*

p.7, last line section 3.3: mesocosm

We corrected the spelling error. Now reads "mesocosm".

p. 8, line 3, paragraph 3: before increasing during

We corrected the spelling error. Now reads "increasing".

p. 3, one but last line, paragraph 3: allochthonous

We corrected the spelling error. Now reads "allochthonous".